# Long Noncoding RNA LINC02470 Sponges MicroRNA-143-3p and Enhances SMAD3-Mediated Epithelial-to-Mesenchymal Transition to Promote the Aggressive Properties of Bladder Cancer

**DOI:** 10.3390/cancers14040968

**Published:** 2022-02-15

**Authors:** Cheng-Shuo Huang, Chen-Hua Tsai, Cheng-Ping Yu, Ying-Si Wu, Ming-Fong Yee, Jar-Yi Ho, Dah-Shyong Yu

**Affiliations:** 1Graduate Institute of Life Sciences, National Defense Medical Center, Taipe 114, Taiwan; qo4m4443151@gmail.com (C.-S.H.); cpyupath@yahoo.com.tw (C.-P.Y.); wuqiqi850119577@gmail.com (Y.-S.W.); 2Graduate Institute of Pathology and Parasitology, National Defense Medical Center, Taipei 114, Taiwan; 3Cheng-Hsin General Hospital, Taipei 112, Taiwan; hua1997@hotmail.com; 4School of Medicine, National Defense Medical Center, Taipei 114, Taiwan; mingfong1998@gmail.com; 5Division of Urology, Department of Surgery, Tri-Service General Hospital, National Defense Medical Center, Taipei 114, Taiwan

**Keywords:** bladder cancer, LINC02470, miR-143-3p, SMAD3, EMT

## Abstract

**Simple Summary:**

Long noncoding RNAs (lncRNAs) were proposed as novel tumor prognostic markers, including for predicting bladder cancer progression, and the competing endogenous RNA (ceRNA) hypothesis conceived an accessible entry point to discover potential lncRNA candidates. This study indicated that LINC02470 promotes bladder cancer cell viability, migration, invasion, and in vivo tumorigenicity by sponging miR-143-3p and consequently rescuing SMAD3 translation to activate the TGF-β-induced EMT process. These data demonstrate that the LINC02470–miR-143-3p–SMAD3 ceRNA axis directly regulates the major transcription factor of TGF-β signaling, SMAD3, thereby inducing the EMT process in bladder cancer and enhancing the aggressiveness of bladder cancer cells.

**Abstract:**

Bladder cancer progression and metastasis have become major threats in clinical practice, increasing mortality and therapeutic refractoriness; recently, epigenetic dysregulation of epithelial-to-mesenchymal transition (EMT)-related signaling pathways has been explored. However, research in the fields of long noncoding RNA (lncRNA) and competing endogenous RNA (ceRNA) regulation in bladder cancer progression is just beginning. This study was designed to determine potential EMT-related ceRNA regulation in bladder cancer progression and elucidate the underlying mechanisms that provoke aggressiveness. After screening the intersection of bioinformatic pipelines, LINC02470 was identified as the most upregulated lncRNA during bladder cancer initiation and progression. Both in vitro and in vivo biological effects indicated that LINC02470 promotes bladder cancer cell viability, migration, invasion, and tumorigenicity. On a molecular level, miR-143-3p directly targets and reduces both LINC02470 and SMAD3 RNA expression. Therefore, the LINC02470–miR-143-3p–SMAD3 ceRNA axis rescues SMAD3 translation upon LINC02470 sponging miR-143-3p, and SMAD3 consequently activates the TGF-β-induced EMT process. In conclusion, this is the first study to demonstrate that LINC02470 plays a pivotally regulatory role in the promotion of TGF-β-induced EMT through the miR-143-3p/SMAD3 axis, thereby aggravating bladder cancer progression. Our study warrants further investigation of LINC02470 as an indicatively prognostic marker of bladder cancer.

## 1. Introduction

Bladder cancer is the seventh most common cancer worldwide and the second most common malignancy of urological organs. Patients who have superficial tumors exhibit initially good responses to transurethral resection and intravesical chemotherapy or immune therapy; however, approximately 60% of patients ultimately experience local recurrence, and up to 40% of patients progress to invasive or metastatic disease with potential lethality [1,2]. A high frequency of relapse and poor clinical outcomes of bladder cancer are associated with tumor progression to muscle invasion, distant metastasis, and therapeutic refractoriness [3,4,5,6]. Therefore, it is necessary to elucidate the mechanisms that underlie bladder cancer progression.

Epithelial-to-mesenchymal transition (EMT) is a multistep process in which epithelial cells lose their epithelial properties and gain mesenchymal properties, such as migration and invasion abilities [7]. Accumulated evidence indicates that EMT is associated with cancer cell invasion and metastasis in various malignancies [8,9]. For the same reason, the EMT process has been found to be comprehensively involved in the relapse, progression, metastasis, and therapeutic refractoriness of bladder cancer [10,11]; however, the detailed regulation of each EMT factor is far from understood. Recently, epigenetic regulation of EMT-related signaling pathways has been addressed on bladder cancer, especially the interactions of noncoding RNAs [12].

Long noncoding RNAs (lncRNA) are noncoding RNAs with lengths exceeding 200 nt that are involved in a variety of cellular physiological processes [13] and participate in bladder cancer initiation, progression, and therapeutic resistance (reviewed in [14,15,16,17]). In addition, microRNAs (miRNAs) function as ubiquitous posttranscriptional regulators that affect RNA stability and translational rate by pairing to miRNA response elements within target RNAs [18,19]. The competing endogenous RNA (ceRNA) hypothesis proposes that there are many interactions between miRNAs and their RNA targets. Therefore, a single miRNA represses multiple RNA targets, and a single RNA target contains multiple MREs for multiple miRNAs; consequently, miRNAs and RNAs regulate each other after competing for a limited pool of miRNAs [20,21]. lncRNAs and ceRNAs have been well documented in EMT-related cancer progression, but the study of EMT-related ceRNAs in bladder cancer progression is just beginning. The ceRNA hypothesis is an easily accessible strategy to discover the interaction between EMT members and lncRNAs or other noncoding RNAs [12]. This study aimed to screen potential ceRNA candidates involved in regulating the EMT process in bladder cancer cells. Our findings demonstrate that the LINC02470–miR-143-3p–SMAD3 ceRNA axis directly regulates the major transcription factor of TGF-β signaling, SMAD3, thereby inducing the EMT process in bladder cancer and enhancing the aggressiveness of bladder cancer cells.

## 2. Results

### 2.1. LINC02470 Is Positively Associated with Bladder Cancer Progression

To identify potential lncRNAs involved in bladder cancer progression, lncRNA screening targets were extracted from the lnCAR database. First, the 400 most distinguished lncRNAs between the tumor vs. normal dataset were selected, with the top 200 upregulated lncRNAs in bladder tumors compared with normal bladder tissues identified as possible oncogene-like lncRNAs, and the top 200 downregulated lncRNAs in bladder tumors compared with normal bladder tissues identified as possible tumor suppressor-like lncRNAs. Second, the intersection among the 400 candidates was analyzed, with the 400 most distinguished lncRNAs between the high grade vs. low grade datasets and the 400 most distinguished lncRNAs between the high stage vs. low stage datasets compared. Only eight candidate lncRNAs were upregulated or downregulated in all three of those datasets (Figure 1A); specifically, MIR4697HG, LINC01451, GLIDR, and LOC728673 were downregulated, while TRMU, IGFL2-AS1, LINC01291, and LINC02470 were upregulated.

Subsequent qRT–PCR validation was performed in cell lines derived from two low-grade (TSGH-8301 and TSGH-9202) and two high-grade (J82 and T24) bladder cancer cases. Only LINC02470 revealed a pattern similar to the screening criterion: it was expressed at lower levels in both low-grade cancer cell lines but at higher levels in both high-grade cancer cell lines (Figure 1B). Moreover, higher LINC02470 expression was associated with worse overall survival of bladder cancer patients in the GSE13507 dataset; LINC02470 conferred a 2.156-fold higher risk (95% CI 1.256–3.700, *p* = 0.004, log-rank test) of bladder cancer-related death (Figure 1C).

### 2.2. LINC02470 Promotes the Aggressive Properties of Bladder Cancer Cells

To investigate the effect of LINC02470 on the aggressiveness of bladder cancer cells, a loss-of-function design was introduced to the two high-grade cell lines, T24 and J82, which exhibit higher endogenous LINC02470 expression. A mixture of three siRNAs targeting LINC02470 (si2470) was employed to avoid low targeting efficiency of a single siRNA type. The efficiency of si2470 was first examined with real-time PCR, and si2470 significantly reduced endogenous LINC02470 in both T24 and J82 cells compared to scrambled controls (sc). In contrast, a gain-of-function design was introduced to the one low-grade cell line, TSGH-8301, which exhibits lower endogenous LINC02470 expression. Ectopic LINC02470 overexpression (2470) significantly increased LINC02470 in TSGH-8301 cells compared to the vector control (Figure 2A). Subsequently, the biological effects of LINC02470 knockdown or overexpression on cell viability and motility were evaluated. Cell viability was significantly inhibited in the LINC02470 knockdown group compared to the sc group but enhanced in the LINC02470 overexpression group compared to the vector group (Figure 2B). This result indicates that LINC02470 is involved in promoting cell-cycle progression in bladder cancer cells. Therefore, LINC02470 regulation of cell-cycle-related molecules was also evaluated using Western blotting. Not only the G1/S transition markers cyclin E1, cyclin E2, and CDK2 but also the survival signaling marker NF-κB were inhibited in cells with LINC02470 knockdown compared to the sc group (Figure 2C,D).

Moreover, cell motility was evaluated via cell migration and invasion assays. In wound-healing assays, knockdown of LINC02470 impaired T24 and J82 cell migration, but overexpression of LINC02470 enhanced TSGH-8301 cell migration (Appendix A). In Transwell assays with or without Matrigel coating, the cell migration and invasion abilities were examined, and knockdown of LINC02470 reduced the number of cells that passed through the chamber compared with the sc group, whereas overexpression of LINC02470 increased the passed-through cell numbers compared with the vector control group (Figure 2E–H). These data indicate that LINC02470 not only participates in the enhancement of cell viability but also aggravates the motility of bladder cancer cells.

### 2.3. LINC02470 Enhanced EMT-Related Signaling Pathways in Bladder Cancer Cells

LINC02470 is involved in signaling that promotes cell viability and motility in bladder cancer, and EMT, as one of the most promising paths, has been comprehensively reported to promote cell proliferation and motility in most cancer types [8,9,22] and has also been reported to be dysregulated and aggravated in bladder cancer progression [10,11]. First, knockdown of LINC02470 significantly downregulated the EMT-TFs SNAIL, SLUG, and ZEB2 compared to the sc groups in both T24 and J82 cells. Knockdown of LINC02470 also reduced the levels of two typical mesenchymal markers, vimentin and N-cadherin, in both bladder cancer cell lines. However, the epithelial marker E-cadherin was not expressed in the LINC02470 knockdown or sc group in either cell line (Figure 3A–C). However, overexpression of LINC02470 significantly upregulated the EMT-TFs SNAIL, SLUG, TWIST1, and ZEB2 compared to the vector group in TSGH-8301 cells (Figure 3A,D). These results indicate that LINC02470 promotes the EMT process, thereby enhancing cell migration and invasion (Figure 2E–H).

Further evaluation of the major EMT upstream signaling pathways was carried out via RT-qPCR, namely, the β-catenin signaling, NOTCH signaling, and SMAD2/3 signaling pathways. SMAD3 was significantly reduced under LINC02470 knockdown in T24 and J82 cells but increased under LINC02470 overexpression in TSGH-8301 cells compared to the relative control groups. However, other representative molecules in each pathway were not altered under knockdown or overexpression of LINC02470, including CTNNB1 and TCF-4, which indicate β-catenin signaling activation, NOTCH1 and NOTCH4, two major dysregulated NOTCH ligands, which induce NOTCH signaling activation in cancer development, and SMAD2 and SMAD4, which induce SMAD2/3 signaling activation (Figure 3E–G). These results indicate that LINC02470 promotes the EMT process by regulating the TGF-β-induced EMT process.

### 2.4. LINC02470 Functions as a Sponge of miR-143-3p to Rescue SMAD3 Expression in Bladder Cancer Cells

It has been an intuitive strategy to identify direct targets of lncRNAs by determining whether they have posttranscriptional interactions with interesting molecules in a ceRNA control model. Bioinformatic screening for ceRNA of LINC02470 was carried out in LncActdb2.0 [23], LnCeVar [24], miRcode [25], and LncBase v3. [26], and the ceRNA unions were subsequently filtered out for EMT-related genes. The LINC02470–miR-143-3p–SMAD3 ceRNA axis was explored as a potential candidate. To verify the relationship among LINC02470, miR-143-3p, and SMAD3, the relative distribution of LINC02470, miR-143-3p, and EMT markers was first compared among different bladder cell lines. LINC02470 and SMAD3 were more highly expressed in high-grade cancer cells with higher mesenchymal-like traits, such as higher N-cadherin and vimentin expression but lower E-cadherin and cytokeratin 18 expression. Conversely, LINC02470 and SMAD3 were expressed at lower levels in low-grade cancer cells or nontumor cells with higher epithelial-like traits, such as higher E-cadherin and cytokeratin 18 expression but lower N-cadherin and vimentin expression. In addition, miR-143-3p exhibited a reverse trend; that is, it was highly expressed in low-grade cancer cells or nontumor cells but more highly expressed in high-grade cancer cells (Figure 4A,B).

Bioinformatic analysis was also performed the coordinated expression patterns between SMAD3–LINC02470, SMAD3–miR-143, and LINC02470–miR-143, and their correlation was further analyzed. The expression of SMAD3 and LINC02470 revealed a positive correlation but the expression of miR-143 was negatively correlated with LINC02470 or SMAD3 in TCGA-BLCA dataset (Appendix A).

After alignment of miR-143-3p, LINC02470, and SMAD3-3′UTR, the typical miR-143-3p binding seed sequence was identified in both LINC02470 and SMAD3-3′UTR (Figure 4C). To confirm the specificity of the miR-143-3p binding seed sequences, the miR-143-3p inhibitory capacity was evaluated with luciferases fused with a wildtype fragment of LINC02470 (LINC02470-WT) or a mutant fragment of the seed sequences. Compared to LINC02470-WT, the LINC02470-mutant lost its inhibitory ability to reach the level of miR-NC controls (Figure 4D). Similarly, miR-143-3p only significantly reduced the activity of the luciferase carrying wildtype SMAD3-3′UTR but lost its inhibitory ability in the luciferase carrying mutant SMAD3-3′UTR, with no difference compared with the miR-NC controls (Figure 4E). Moreover, when transcription was blocked by actinomycin D, both LINC02470 (Figure 4F) and SMAD3 (Figure 4G) RNA was degraded more quickly over time in miR-143-3p-transfected T24 cells than in miR-NC controls. These results not only indicate that miR-143-3p directly targets and degrades SMAD3 mRNA but also suggest that LINC02470 sponges miR-143-3p; that is, LINC02470-miR-143-3p-SMAD3 forms a ceRNA axis that might result in degradation of either RNA molecule.

Moreover, T24 cells transfected with miR-143-3p mimic exhibited lower LINC02470 and SMAD3 mRNA expression than miR-NC control cells (Figure 5A,E left). In contrast, T24 cells transfected with miR-143-3p inhibitor showed higher LINC02470 and SMAD3 mRNA expression than those transfected with the miR-NC inhibitor control (Figure 5B,F left). To further evaluate the interaction between LINC02470 and miR-143-3p, T24 cells were co-transfected with either si2470 or sc combined with miR-143-3p mimic or miR-143-3p inhibitor and compared to those transfected with either si2470 or sc combined with miR-NC or miR-NC inhibitor (NCI) control. Cells co-transfected with si2470 and miR-143-3p mimic showed the highest inhibitory patterns of SMAD3 protein expression and its active phosphorylated form, along with the EMT-TFs SNAIL and SLUG and the mesenchymal effector N-cadherin (Figure 5C,E right), compared to si2470 combined with miR-NC control and sc combined with miR-NC or miR-143-3p mimic controls (Figure 5E left). Conversely, cells co-transfected with si2470 and miR-143-3p inhibitor exhibited compensated expression patterns in which SMAD3 protein, active p-SMAD3, SNAIL, SLUG, and N-cadherin expression levels were rescued in si2470-transfected cells after ablation of endogenous miR-143-3p (Figure 5D,F right) compared to si2470 combined with miR-NCI control and sc combined with miR-NC or miR-143-3p inhibitor controls (Figure 5F left). These results confirm that LINC02470 and miR-143-3p bound to and inhibited each other, effectively regulating the SMAD3-related EMT process.

Moreover, the miR143-3p mimics significantly inhibited SMAD3, active phosphorylated SMAD3, SLUG, and N-cadherin. Furthermore, overexpressed SMAD3 rescued the inhibitory effects of miR-143-3p mimics and recovered the expression of SMAD3, active phosphorylated SMAD3, SLUG, and N-cadherin (Appendix A).

### 2.5. Knockdown of LINC02470 Reduced Tumorigenicity In Vivo

To further illustrate the role of the LINC02470–miR-143-3p–SMAD3 ceRNA axis in bladder tumorigenicity in vivo, transfected T24 cells were subcutaneously injected into the flank of the left hind leg of 8 week old nude mice. The tumor sizes in the shLINC02470 (sh2470) group were significantly smaller than those in the shRNA negative control (shNC) group (Figure 6A). In addition, the tumor growth rate and tumor weight in the sh2470 group were significantly reduced compared to those in the shNC control group (Figure 6B,C). The body weights in both groups were not affected by different cell inoculations (Figure 6D).

Upon further comparison of protein expression in FFPE tissues from both groups using IHC staining, SMAD3 was revealed to have a higher expression level in the shNC group than in the sh2470 group (Figure 6E). Moreover, using RT-qPCR, the LINC02470 expression level was found to be higher in the shNC group than in the sh2470 group. In contrast, miR-143-3p was more highly expressed in the sh2470 group than in the shNC group (Figure 6F). These results indicate that knockdown of LINC02470 conferred suppressive effects on tumorigenicity in vivo by downregulating the SMAD3-induced EMT process.

## 3. Discussion

Cancer progression and metastasis have become major threats in clinical practice to increase cancer mortality and therapeutic refractoriness, including in bladder cancer, and dysregulation of the EMT process has been comprehensively studied and found to be involved in these cancer development stages [8,9]. In the current study, our results indicated that the LINC02470–miR-143-3p–SMAD3 ceRNA axis is involved in bladder cancer progression by activating the SMAD3-induced EMT process (a schematic diagram is illustrated in Figure 7). Moreover, SMAD3 activates EMT-TFs, such as SNAIL and SLUG, to provoke mesenchymal transition and ultimately enhance cancer cell viability and motility. There are two novel findings in the current study. First, the LINC02470–miR-143-3p–SMAD3 ceRNA axis participates in bladder cancer progression. This is also the first study to elucidate the molecular regulation of LINC02470. Second, miR-143-3p fluctuation significantly altered the EMT process by regulating EMT-TF expression, thereby modifying epithelial-to-mesenchymal traits.

To date, this is the first connection identified between the LINC02470–miR-143-3p–SMAD3 axis and bladder cancer progression. Our previous study reported that exosomal LINC02470 promotes EMT and aggressiveness of bladder cancer cells, but the underlying mechanism was unclear. In this study, we employed the ceRNA hypothesis to investigate interactions of LINC02470 with miRNA and mRNA and found that LINC02470 bound to miR-143-3p, resulting in degradation of either RNA molecule [27,28]. Consequently, LINC02470 competed with miR-143-3p for the SMAD3-3′UTR and rescued SMAD3 translation. SMAD3 subsequently induced EMT-TF expression and induction of more mesenchymal traits, including higher cell viability and higher cell migration and invasion capacity.

In addition, miR-143 is a comprehensively reported tumor suppressor miRNA and has been reported to directly bind and inhibit the translation of several oncogenes (reviewed in [29,30,31]). Therefore, downregulation of miR-143 has been described in a considerable number of cancer cell lines and tumors [31], including bladder cancer [32,33,34], and miR-143 usually exhibits an expression pattern similar to that of its evolutionarily conserved microRNA cluster partner miR-145 [35].

The miR-143/-145 cluster has been reported to be reduced in different tumors through various mechanisms [31], such as genetic deletion in myelodysplastic syndrome [36] and ovarian carcinoma [37], as well as epigenetic silencing via CpG island methylation in prostate [38], lung [39], and esophageal squamous cell carcinomas [40]; moreover, downregulation of several tumor suppressor genes or upregulation of certain oncogenes may also indirectly reduce miR-143 and miR-145 expression. Some double-negative feedback loops, including miR-143 (LIMK1, SOX2) and miR-145 (ADAM17, NEDD9, SOX2), also result in miR-143 and miR-145 downregulation in tumor cells (reviewed in [35]). In the present study, we report that the interaction between LINC02470–miR-143-3p is a rising novel posttranscriptional regulation that might also result in degradation of both RNA molecules. Recently, potential roles of miR-143 in ceRNA regulation were also noticed, and several other lncRNA–mRNA pairs have been reported in bladder cancer, including the PCAT6–miR-143-3p–PDIA6 axis [41], LINC00511–miR-143-3p–PCMT1 axis [42], SNHG1–miR-143-3p–EZH2 axis [43], MAFG–AS1–miR-143-3p–COX-2 axis [44], circ_0006332–miR-143–MYBL2 axis [45], FOXD2–AS1–miR-143–ABCC3 axis [46], and UCA1–miR-143–HMGB1 axis [47]. This is also the first study to address miR-143(-3p)-regulated ceRNA in EMT process regulation.

Several major components of the TGF-β pathway have been proposed to be directly targeted by miR-143 or miR-145, indicating that they may constitute a negative feedback loop. Among them, miR-145 directly targets SMAD3, a key transcription factor involved in TGF-β responses, which has been confirmed in lung cancer [48]. Similarly, downregulation of miR-145 corresponded to higher SMAD3 expression and EMT process induction in nasopharyngeal cancer cells [49] or inhibited TGFβR2 and SMAD3 in bladder cancer [50]. In the current study, our findings are the first to demonstrate that miR-143-3p directly targets and inhibits SMAD3 expression, thereby potentially strengthening the negative feedback loop upon cooperation with miR-145.

TGF-β is one the most well-characterized inducers of the miR-143/-145 cluster, which is often induced during tumorigenesis or tumor progression through transcriptional and posttranscriptional regulation of SMAD factors [51,52,53]. TGF-β acts as a double-edged sword in cancer that initially suppresses tumorigenesis via its antiproliferative properties but drives tumor progression and metastasis via its strong ability to induce EMT [54,55,56]. Interestingly, there is a main conventional SMAD2/3/4 binding consensus element (CAGAC motif) located on the promoter of LINC02470 (Figure 7B). This explains how both miR-143/-145 anti-oncogenic properties cooperate with the TGF-β antiproliferative ability to suppress tumorigenesis in the early stage, but TGF-β provokes tumor progression at the late stage by inducing lncRNAs to compete against and break down miR-143-3p in a negative feedback loop. Our results indicated that linc02470 is a potent ceRNA competitor of miR-143-3p, although other ceRNA competitors of miR-143 or miR-145 still need to be explored in the future. Dysregulation of the EMT process has become a major issue in bladder cancer progression and therapeutic responses [10,11], and lncRNAs or other ncRNAs participate in important regulatory processes (reviewed in [14,15,16,17]). The ceRNA hypothesis has been employed as an easily accessible strategy to discover interactions between EMT members and lncRNAs or other ncRNAs [12]. Although several ceRNA pairs have been reported to regulate the EMT process in bladder cancer, the present study may shed light on the fact that the LINC02470–miR-143-3p–SMAD3 axis forms feedback loops with TGF-β at different tumor stages, resulting in dual functions, which may help to explain the “TGF-β paradox”, i.e., the dichotomous nature of TGF-β during tumorigenesis.

SMAD3 is a key transcription factor member of the TGF-β signaling family, and it cooperates with other SMADs to activate the expression of downstream EMT-related genes, such as SNAIL and SLUG [57]. Upon genome-wide expression profile screening, high expression of SMAD3 was found to be significantly associated with worse overall survival and higher tumor invasive depth in bladder cancer [58]. SMAD3/SMAD4 sufficiently activates TGF-β receptors and enhances EMT induction [59]. Certain miRNAs have been reported to target and reduce SMAD3 expression, thereby inhibiting the EMT process in bladder cancer cells, including the miR-665–SMAD3 axis [60], miRNA-145–TGFBR2–SMAD3 axis [50], and miR-323a-3p–MET–SMAD3 axis [61], and only one ceRNA profiling was employed to identify the PlncRNA-1–miR-136–SMAD3 axis [62]. Additionally, indirect suppression of upstream TGF-β signaling also verified that SMAD3 is necessary to promote EMT in bladder cancer cells [63,64,65]. The findings of the present study indicate that miR-143-3p directly targets and suppresses SMAD3 expression to inhibit the EMT process in bladder cancer cells; however, high LINC02470 expression sponges miR-143-3p to rescue SMAD3 translation and recovers the EMT process.

## 4. Materials and Methods

### 4.1. Cell Lines, Plasmids, and Transfection

Six human bladder cancer cell lines (low-grade cancer: TSGH-8301, TSGH-9202, RT4, and HT-1376; and high-grade cancer: T24 and J82) were originally acquired from the ATCC or the Bioresource Collection and Research Center. All cells were incubated in RPMI 1640 medium containing 10% fetal bovine serum, 1 μg/mL penicillin, and 1 μg/mL streptomycin (Life Sciences, Palo Alto, CA, USA) at 37 °C in a 5% CO_2_ humidified incubator.

GPU6/GFP/Neo-LINC02470-Homo794 (sh02470) and pGPU6/FGP/Neo-shNC (shNC) were purchased from GenePharma (Shanghai, China), whereas pCMV6-SMAD3 plasmid, miRNA mimic of miR-143-3p, miR-NC control, miR-143-3p inhibitor, and miR-inhibitor control were purchased from Sino Biologic, Inc. (Beijing, China). pCMV-LINC02470 (2470) and pCMV vector (vector) control were purchased from Addgene. For transfection with sh02470, shNC, 2470, pCMV6-SMAD3, and vector plasmids, cells were transfected with 2 µg of plasmids for 48 h and harvested for analysis, or 400 µg/mL G418 was added for 1 month to obtain stable clones for xenografts. For siRNA and miRNA experiments, cells were transfected with 100 pmol small RNA for 48 h and harvested for analysis.

### 4.2. Bioinformatic Dataset Analysis

Bioinformatic data of lncRNA screening were obtained from lnCAR. ceRNA prediction was performed on the websites LncActdb2.0, LnCeVar, miRcode, and LncBase v3. In addition, GEO data were analyzed using lnCAR. The correlations between SMAD3 and LINC02470 between, miR-143 and LINC02470, or between miR-143 and SMAD3 were analyzed with TCGA-BLCA dataset on the kmplot.com (accessed on 2 January 2022). The LINC02470 data were collected with discrete percentage and analyzed with Spearman rho correlation, whereas miR-143 and SMAD3 were recorded as continuous variables and analyzed with Pearson correlation.

### 4.3. RT-qPCR Assay

Total RNA was extracted from cultured cells using TOOLSmart RNA Extractor (BIOTOOLS, Taiwan). The concentration of total RNA was evaluated using a NanoDrop spectrophotometer (Thermo Fisher Scientific, Waltham, MA, USA). Total RNA was reverse-transcribed into cDNA using a ToolsQuant II Fast RT Kit with Oligo (dT) primer (BIOTOOLS, Taiwan) in a 20 μL reaction system consisting of 1000 ng of template, 2 μL of 10× RT Reaction Premix with Oligo (dT) primer, 1.5 μL of ToolsQuant II Fast RT, and RNase-free ddH_2_O. The mixture was centrifuged briefly and incubated for reverse transcription at 42 °C for 15 min, followed by enzyme inactivation at 85 °C for 5 min. Quantitative real-time PCR was performed in a 20 μL reaction system containing 3 μL of diluted cDNA, 10 μL of TOOLS 2× SYBR qPCR Mix (BIOTOOLS, Taiwan), 0.5 μL of gene-specific forward and reverse primers (10 pmol/μL), and 6.5 μL of RNase-free ddH_2_O on a QuantStudio 5 Real-Time PCR System (Applied Biosystems, Foster City, CA, USA) according to the manufacturer’s instructions. Appendix A lists the primer sequences used in this study. Briefly, after an initial denaturation step at 95 °C for 15 min, amplifications were carried out for 40 cycles at a melting temperature of 95 °C for 15 s, an annealing temperature of 60 °C for 20 s, and an elongation temperature of 72 °C for 20 s. The specificity of amplicons was confirmed by melting curve analysis. GAPDH was used as a reference gene. The relative expression levels of target genes were calculated using the 2^−ΔΔCT^ method. All experiments were conducted in triplicate, and no-template controls were included in each run.

### 4.4. Immunohistochemistry (IHC)

IHC was performed in formalin-fixed paraffin-embedded (FFPE) specimens. Each 4 μm dissection slide was blocked with 10% goat serum for 1 h and incubated with SMAD3 antibody (Cell Signaling Technology, cat. No.#9513; Beverly, MA, USA) for 2 h at room temperature. After washing three times in TBST (10 mM Tris pH 7.4, 150 mM NaCl, 0.1% Tween-20) for 10 min, slides were processed according to the Super Sensitive Polymer HRP Detection System/DAB kit instructions (Thermo Scientific, Waltham, MA, USA), counterstained with hematoxylin, and imaged with a microscope (Leica Microsystems, Mannheim, Germany).

### 4.5. Plasmid Construction and Luciferase Reporter Assay

The human LINC02470 miR-143-3p binding fragment and the SMAD3 3′UTR containing the miR-143-3p binding site were amplified via PCR from human genomic DNA using the primers listed in Appendix A and were inserted between the *Spe*I/*Hind*III sites in a pMIR-reporter vector (Invitrogen, Waltham, MA, USA). All constructs were verified by autosequencing.

A total of 5000 T24 cells were seeded into a 24-well plate 16 h before transfection. For the miR-143-3p binding assay, cells were co-transfected with 120 pmol of miR-143-3p mimic or miR-NC control and with 1 µg of pMIR-reporter empty vector or pMIR-reporter carrying the wildtype or mutant LINC02470 miR-143-3p binding fragment or SMAD3-3’UTR using Lipofectamine 2000 (Invitrogen). In addition, cells were co-transfected with 500 ng of β-gal control vector for each combination as an internal control. Luciferase assays were performed with a Dual-Light Luciferase and β-Galactosidase Reporter Gene Assay System (Invitrogen, Waltham, MA, USA) according to the manufacturer’s instructions. Luminescence was detected using a SpectraMax i3x (Molecular Devices LLC, San Jose, CA, USA), and six replicates were prepared for each condition.

### 4.6. Western Blotting Assay

Total protein was extracted using RIPA buffer (Thermo Fisher Scientific, Waltham, MA, USA) supplemented with a protease inhibitor cocktail (Roche, Basel, Switzerland) at a ratio of 100:1. The protein concentration was determined with a BCA protein assay kit (Thermo Fisher Scientific, Waltham, MA, USA). Proteins (30 µg) were subjected to 10% SDS-PAGE and were then transferred onto a polyvinylidene fluoride (PVDF) membrane (Millipore, Burlington, MA, USA). The membrane was blocked with 5% BSA (Sigma–Aldrich, Burlington, MA, USA) in TBST (10 mM Tris pH 7.4, 150 mM NaCl, 0.1% Tween-20) and incubated with primary antibodies against SMAD3 (Cell Signaling Technology, cat. no.#9513; Beverly, MA, USA), p-SMAD2/3 (Cell Signaling Technology, cat. no.#8828; Beverly, MA, USA), SNAIL (Abcam, cat. no. ab53519; Cambridge, UK), SLUG (Abcam, cat. no. ab27568; Cambridge, UK), TWIST1 (Abcam, cat. no. ab50887; Cambridge, UK), ZEB2 (Cell Signaling Technology, cat. no.#97885; Beverly, MA, USA), E-cadherin (Abcam, cat. no. ab231303; Cambridge, UK), N-cadherin (Abcam, cat. no. ab18203), vimentin (Abcam, cat. no. ab45939; Cambridge, UK), or GAPDH (Cell Signaling Technology, cat. no.#5174; Beverly, MA, USA) overnight at 4 °C. Subsequently, the blots were washed with TBST, followed by incubation with an HRP-conjugated goat anti-mouse or goat anti-rabbit secondary antibody (1:5000; Santa Cruz Biotechnology; Dallas, TX, USA) at room temperature for 1 h. The immunoreactive bands were visualized with Immobilon Western Chemiluminescent HRP Substrate (Millipore, Burlington, MA, USA) and analyzed with the UVP GelStudio PLUS System (Analytik Jena AG, Thuringia, Germany). Each condition was prepared in triplicate. Whole blots (uncropped blots) showing all the bands with all molecular weight markers on the Western are also enclosed in the Appendix A.

### 4.7. Cell Viability Assay

The viability of bladder cancer cells was determined using a 3-[4,5-dimethylthiazol-2-yl]-2,5-diphenyl-tetrazolium bromide (MTT, Sigma–Aldrich, Burlington, MA, USA) assay. Cells were seeded in 96-well plates at 3000 cells/well and cultured overnight. After treatment under the indicated conditions for 48 h, cells were incubated with 0.1 mg/mL MTT for 3 h, and formazan was dissolved in dimethyl sulfoxide (Sigma–Aldrich, Burlington, MA, USA) at room temperature for 10 min. The absorbance at 560 nm was measured with a spectrophotometer (Bio–Rad Inc, Hercules, CA, USA). All experiments were performed in triplicate.

### 4.8. Cell Migration and Invasion Assay

In wound healing assays, 1 × 10^5^ cells were seeded in six-well plates and incubated to 90% confluence before transfection. After treatment under the indicated conditions for 24 h, cells were scraped with a sterile 200 μL pipette tip to generate a clear line in the wells at time 0. The migrated cells were observed with a phase-contrast microscope every 8 h (Leica DMI4000B, Bucks, UK), and the wound width at the designated time was measured with ImageJ software. All experiments were performed in triplicate.

Transwell migration assays were performed using 8 μm Transwell chambers (Corning, Steuben County, NY, USA) with 1 × 10^4^ cells for each treatment. Transwell invasion assays were evaluated with the same chambers coated with 1 mg/mL Matrigel (BD Biosciences, Franklin Lakes, NJ, USA) with 2 × 10^4^ cells for each treatment. The migration and invasion chambers were incubated in a humidified 5% CO_2_ incubator at 37 °C for 24 h. Cells were then fixed with 4% paraformaldehyde, and the inner surface of the upper chambers was wiped with cotton swabs to remove unmigrated or uninvaded cells. After washing, the chambers were stained with crystal violet (Sigma–Aldrich, Burlington, MA, USA) for 15 min, and the Transwell membranes were torn and kept on slides. For each treatment, five random fields were photographed at 100× magnification, and the crystal violet-stained area was calculated using ImageJ software. Each condition was plated in triplicate.

### 4.9. Actinomycin D Assay

T24 cells were treated with 2 mg/mL actinomycin D (Merck, Darmstadt, Germany) to block transcription. Then, the remaining RNAs extracted from treated cells were assessed via qRT–PCR.

### 4.10. Xenograft Tumor Assay

BALB/c nude mice (7 weeks old) were purchased from the National Laboratory Animal Center in Taiwan and acclimated for 1 week. In addition, 1 × 10^7^ T24 cells transfected with shLINC02470 or shNC control were suspended in 100 µL of 1 × PBS with 10 mg/mL Matrigel (BD Biosciences) and implanted into the flank of the left hind leg for each recipient mouse (*n* = 5, each group). Before xenografting, cells were tested for mycoplasma using an e-Myco Mycoplasma PCR Detection Kit (iNtRON Biotechnology, Inc., Seongnam, Korea). The tumor size was measured every 3 days using calipers and was calculated by determining (length × width^2^)/2 until the mice were sacrificed by CO_2_ inhalation. Tumors were removed at Day 60 to measure tumor weight, and tumors were fixed in formalin followed by paraffin embedding and dissection for immunohistochemistry. All experimental and animal care procedures were approved by the Laboratory Animal Center of National Defense Medical Center.

### 4.11. Statistical Analysis

All statistical analyses were performed with SPSS 22.0 (IBM, SPSS, Chicago, IL, USA) and GraphPad Prism 8.0 (GraphPad Software, La Jolla, CA, USA). Data were recorded as continuous variants and analyzed with Student’s *t*-test. The results are expressed as the mean ± standard deviation of at least three separate experiments. All statistical tests and *p*-values were two-sided, and the level of significance was set to <0.05 (*), <0.01 (**), or <0.001 (***).

## 5. Conclusions

In conclusion, this is the first study to demonstrate that LINC02470 plays a significant regulatory role in EMT promotion through the miR-143-3p–SMAD3 signaling pathway, thereby aggravating bladder cancer progression.

## Figures and Tables

**Figure 1 cancers-14-00968-f001:**
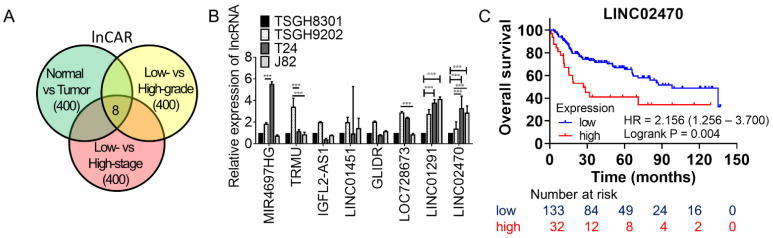
Eight candidate lncRNAs in the lnCAR database were screened out, with subsequent evaluation of LINC02470. (**A**) Eight candidate lncRNAs were obtained from the lnCAR database by intersection of higher expression in bladder cancer than normal specimens, higher expression in high-grade than low-grade specimens, and higher expression in high-stage than low-stage specimens. (**B**) Comparison of MIR4697HG, TRMU, IGFL2-AS1, LINC01451, GLIDR, LOC728673, LINC01291, and LINC02740 expression among different bladder cancer cells via qRT–PCR. (*** *p* < 0.001, Student’s *t*-test) (**C**) Higher LINC2470 expression was associated with worse overall survival of bladder cancer patients in the GSE13507 dataset.

**Figure 2 cancers-14-00968-f002:**
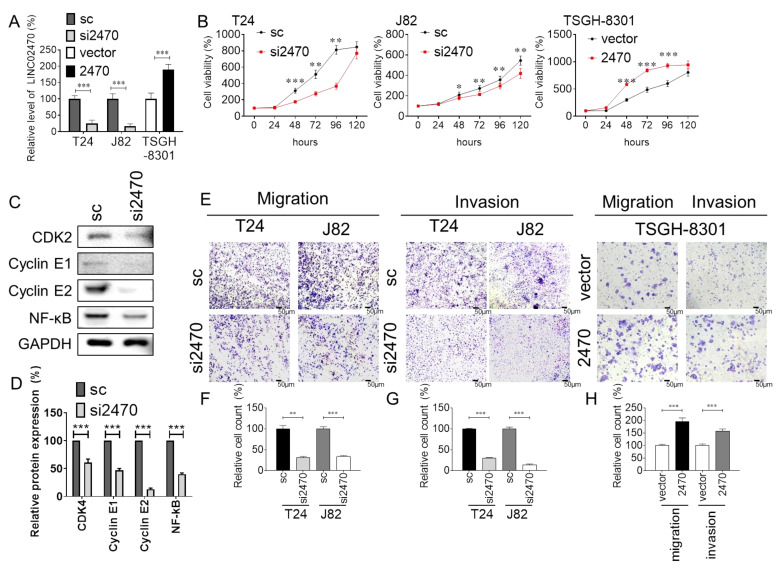
(**A**) The relative expression levels of LINC02470 in T24 and J82 cells transfected with LINC02470 siRNAs (si2470) or scrambled negative control (sc) and in TSGH-8301 cells transfected with pCMV-LINC02470 (2470) or pCMV vector (vector) control (normalized to GAPDH expression). (**B**) Relative cell viability at 0 h, 24 h, 48 h, 72 h, 96 h, and 120 h was compared between T24 and J82 cells transfected with si2470 or sc and between TSGH-8301 cells transfected with 2470 or vector. (**C**) The protein levels of the G1-S transition markers CDK2, cyclin E1, and cyclin E2 and the survival signaling marker NF-κB were determined using Western blotting in T24 cells transfected with si2470 or sc. (**D**) Bar chart showing the relative protein levels, which were normalized to GAPDH; the sc group was used as the comparative baseline. (**E**) LINC02470 effects on cell migration and invasion abilities were analyzed with Transwell assays. T24 and J82 cells transfected with si2470 exhibited fewer migrated and invaded cells than the sc group. (**F**–**H**) Bar charts showing the relative migrated/invaded cell numbers (* *p* < 0.05, ** *p* < 0.01, *** *p* < 0.001, Student’s *t*-test).

**Figure 3 cancers-14-00968-f003:**
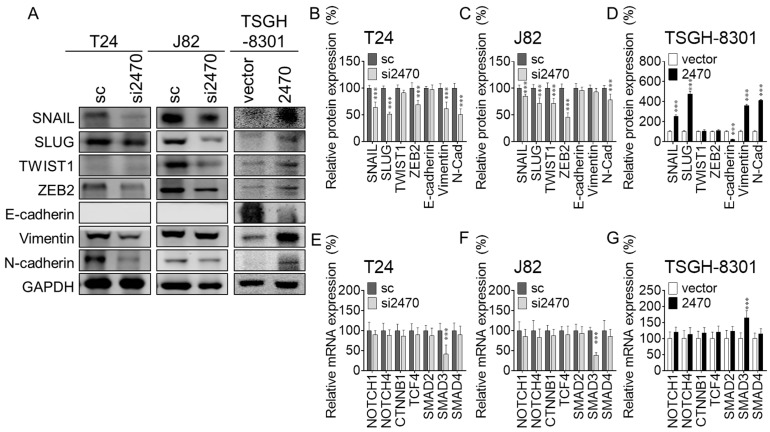
(**A**) The protein expression of EMT-related downstream signaling molecules was compared between the si2470 and sc groups in T24 and J82 cells or the 2470 and vector groups in TSGH-8301 cells. The EMT-TFs SNAIL, SLUG, and ZEB2 and the mesenchymal effectors vimentin and N-cadherin were significantly reduced in the si2470 group compared to the sc group in T24 and J82 cells. SNAIL, SLUG, TWIST1, and ZEB2 and the mesenchymal effectors vimentin and N-cadherin were increased but the epithelial effector E-cadherin was reduced in the 2470 group compared to the vector group in TSGH-8301 cells. (**B**–**D**) Bar chart showing the relative protein levels, which were normalized to GAPDH; the sc group was used as the comparative baseline. (**E**–**G**) The major EMT-upstream signaling pathways were compared via RT–qPCR, namely, the β-catenin signaling (CTNNB1, TCF4), NOTCH signaling (NOTCH1, NOTCH4), and SMAD2/3 signaling (SMAD2, SMAD3, SMAD4) pathways. SMAD3 was significantly reduced in si2470-transfected T24 or J82 cells compared to the sc groups but increased in 2470-transfected TSGH-8301 cells compared to the vector group. However, other selected molecules in each pathway were not altered under knockdown or overexpression of LINC02470 (*** *p* < 0.001, Student’s *t*-test).

**Figure 4 cancers-14-00968-f004:**
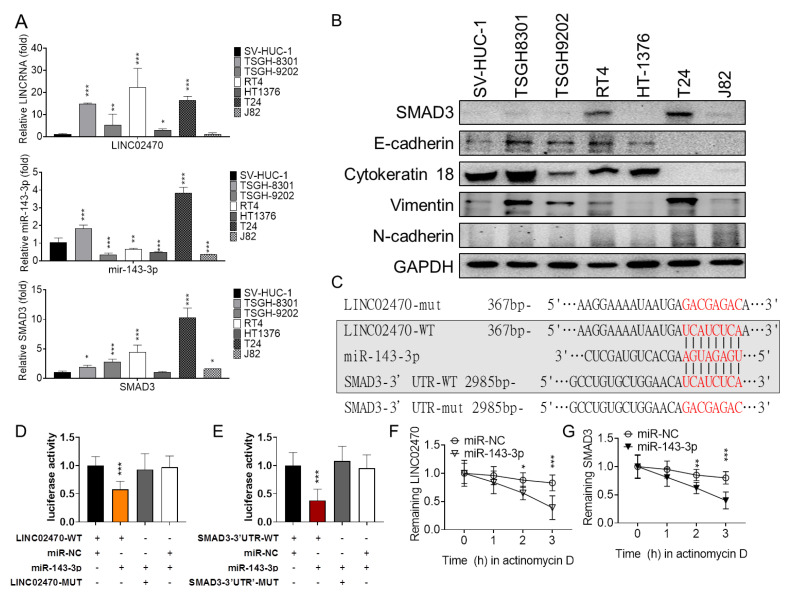
(**A**) The relative RNA expression levels of LINC02470, miR-143-3p, and SMAD3 mRNA were compared among bladder cell lines: the nontumorigenic urothelial cell line SV-HUC-1, the low-grade bladder cancer cell lines TSGH-8301, TSGH-9209, RT4, and HT-1376, and the high-grade bladder cancer cell lines T24 and J82. (**B**) The relative protein expression levels of SMAD3, the epithelial effectors E-cadherin and cytokeratin 18, and the mesenchymal effectors vimentin and N-cadherin were compared among the same series of cells described above. (**C**) The predicted miR-143-3p targeting site within the LINC02470 and SMAD3 3′UTR and the mutated sites were aligned as indicated. (**D**) After co-transfection with miR-143-3p mimic or miR-NC, the relative luciferase activity of a pMIR-reporter carrying wildtype or mutant LINC02470 was determined in T24 cells and normalized to the luciferase activity in the LINC02470-WT combined with miR-NC transfectants. Cells were also co-transfected with a β-galactosidase control vector as a transfection loading control. miR-143-3p mimic reduced luciferase activity in cells transfected with the pMIR-reporter carrying wildtype LINC02470 compared to miR-NC-transfected cells. (**E**) After co-transfection with the miR-143-3p mimic or miR-NC, the relative luciferase activity of the pMIR-reporter carrying the wildtype or mutant SMAD3 3′UTR was determined in T24 cells and normalized to the luciferase activity in SMAD3-3′UTR-WT combined with miR-NC transfectants. Cells were also co-transfected with a β-galactosidase control vector. miR-143-3p mimic reduced the luciferase activity in cells transfected with the pMIR-reporter carrying the wild-type SMAD3 3′UTR compared to miR-NC-transfected cells. There was no change in luciferase activity in cells co-transfected with miR-143-3p mimic or mNC and the pMIR-reporter carrying a mutant SMAD3 3′UTR. Actinomycin D assays revealed that miR-143-3p mimic accelerated (**F**) LINC02470 lncRNA and (**G**) SMAD3 mRNA degradation compared to the miR-NC groups (* *p* < 0.05, ** *p* < 0.01, *** *p* < 0.001, Student’s *t*-test).

**Figure 5 cancers-14-00968-f005:**
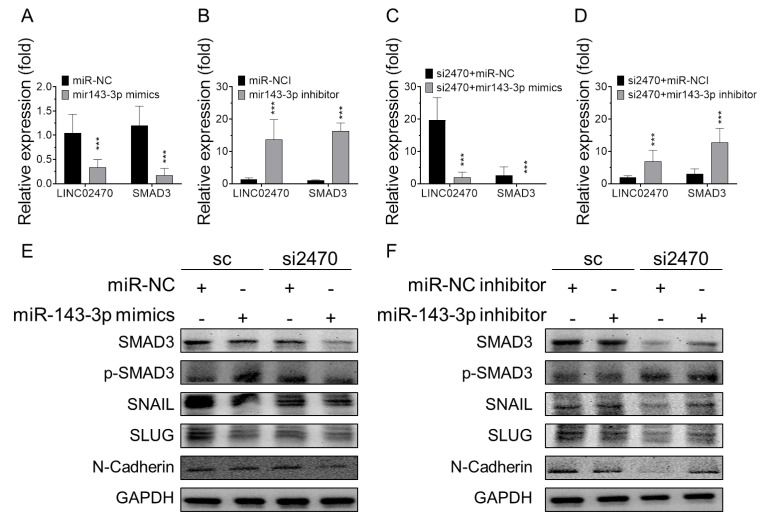
(**A**) LINC02470 and SMAD3 RNA expression levels determined by RT-qPCR were compared in T24 cells transfected with (**A**) miR-143-3p mimic compared to miR-NC control, (**B**) miR-143-3p inhibitor compared to miR-NC inhibitor (NCI) control, (**C**) LINC02470 siRNA combined with miR-143-3p mimic compared to miR-NC control, or (**D**) LINC02470 siRNA combined with miR-143-3p inhibitor compared to NCI control (*** *p* < 0.001, Student’s *t*-test). (**E**,**F**) Protein expression levels of SMAD3 and its downstream EMT markers were compared via Western blotting among the same treated groups.

**Figure 6 cancers-14-00968-f006:**
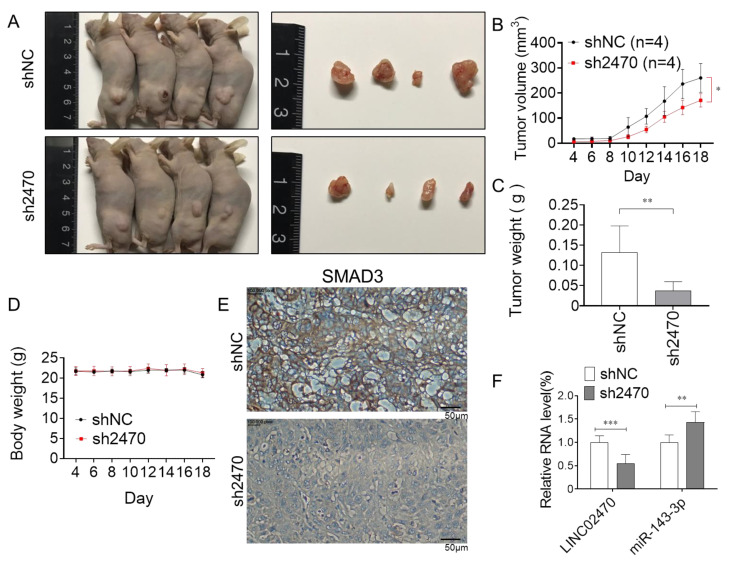
The effect of LINC02470 on the tumorigenicity of bladder cancer cells (T24 cells). (**A**) Representative xenograft tumors and dissected tumor masses of shLINC02470 (sh2470)- and shRNA negative control vector (shNC)-transfected T24 cells are shown. (**B**) Tumor volumes in the sh2470 and shNC group were measured for 18 days, and shLINC02470 reduced the tumor growth rate. (**C**) Tumor weights of tumor masses dissected from the sh2470 and shNC groups were measured. (**D**) Mouse body weight was also measured for 18 days, and sh2470 and shNC did not affect body weight. (**E**) Representative IHC results showing SMAD3 expression in xenograft tumor specimens in the sh2470 and shNC group. (**F**) Relative RNA levels of LINC02470 and miR-143-3p were compared between the sh2470 and shNC groups via RT-qPCR or stem-loop RT-qPCR (* *p* < 0.05, ** *p* < 0.01, *** *p* < 0.001, Student’s *t*-test).

**Figure 7 cancers-14-00968-f007:**
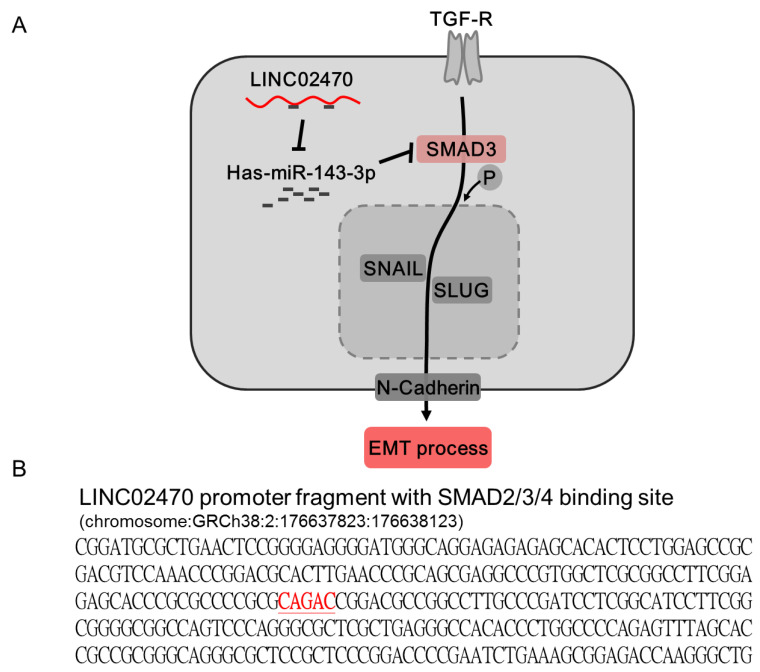
(**A**) Schematic diagram of the LINC02470–miR-143-3p–SMAD3 axis in bladder cancer cells. LINC02470 functions as a miRNA sponge to inhibit miR-143-3p and rescue SMAD3. Subsequently, SMAD3 activates the TGF-β-related EMT process and ultimately aggravates the malignant behaviors of bladder cancer cells. Therefore, LINC02470 plays an oncogenic role during bladder cancer progression. (**B**) SMAD2/3/4 binding consensus motif in the LINC02470 promoter fragment.

## Data Availability

Data are contained within the article.

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
