# Peer review of "Long Noncoding RNA LINC02470 Sponges MicroRNA-143-3p and Enhances SMAD3-Mediated Epithelial-to-Mesenchymal Transition to Promote the Aggressive Properties of Bladder Cancer"

_cancers, 2022, doi:10.3390/cancers14040968_

Round 1

Reviewer 1 Report

The revised manuscript is suitable for publication

Author Response

Thanks for your comment.

Reviewer 2 Report

The authors have performed additional analyses/experiments as suggested. This reviewer has a few more concerns about the rescusing experiment. First, the Western blo results did not clearly show that miR-143-mediated downregulation of SLUG and N-Cad was reversed by SMAD3 overexpression. Quantitative representation of the immunreactive signals on the blots should be provided. Second, evidence for the functional connection between LINC02470 and SMAD3 is critical, which can be demonstrated by LINC knockdown plus SMAD3 overexpression. The authors should include this result in their work. 

Author Response

Thanks for your suggestion. The quantitative results of Figure S3 have been supplemented. And si2470 indeed reduced SMAD3 mRNA expression in both T24 and J82 cells but overexpression LINC02470 rescues SMAD3 mRNA expression in TSGH-8301 cells (Figure 3 E-G). Similarly, si2470 also reduced SMAD3 protein expression so did p-SMAD3, Slug and N-cadherin expression in T24 cells. Oppositely, miR-143-3p inhibitor combined si2470 compensate si2470 inhibitory effect on SMAD3 and rescue p-SMAD3, SLUG and N-Cadherin expression. Those results deduced that there was functional connection between LINC02470 and SMAD3 to form a LINC02470-miR-143-3p-SMAD3 axis in bladder cancer progression.

Reviewer 3 Report

I am satisfied with the revised manuscript.

Author Response

Thanks for your comment.

This manuscript is a resubmission of an earlier submission. The following is a list of the peer review reports and author responses from that submission.

Round 1

Reviewer 1 Report

Dysregulation of epithelial-to-mesenchymal transition (EMT)-related signaling pathways have been shown to be involved in Bladder cancer progression and metastasis. In addition, it has been suggested that long noncoding RNA (lncRNA) and competing endogenous RNA (ceRNA) regulation is important in bladder cancer progression. In this study, the authors identified LINC02470 as the highest upregulated lncRNA during bladder cancer initiation and progression. Both in vitro and in vivo biological effects indicated that LINC02470 promotes bladder cancer cell viability, migration, invasion and tumorigenicity. Furthermore, the authors found that miR-143-3p directly targets and reduces both LINC02470 and SMAD3 RNA expression. Based on their findings, the authors proposed the model that the LINC02470-miR-143-3p-SMAD3 ceRNA axis rescues SMAD3 translation upon LINC02470 sponging miR-143-3p and SMAD3 consequently activates TGF-β-induced EMT process. Overall, these results are interesting and convincing, however a few points should be changed or addressed.

(1) Some of the western blot datas (Snail blots in Figure 3A, SMAD3 and E-cadherin blots in Figure 4B and Figure 5E and F) are enhanced to see the difference. It would be better to show the western blots data not so enhanced in the figures.

(2) The author concluded that LINC02470 sponges miR-143-3p and SMAD3 consequently activates TGF-β-induced EMT process. Is it possible that knockdown of linc02470 decreases SMAD3 interaction with the promoter or enhancer of the EMT related genes including snail or slug? Could the author show some of ChIP (chromatin immunoprecipitation) data, it would be helpful to understand the model shown in Figure 7A.

Author Response

Reviewer 1

Q1. Some of the western blot datas (Snail blots in Figure 3A, SMAD3 and E-cadherin blots in Figure 4B and Figure 5E and F) are enhanced to see the difference. It would be better to show the western blots data not so enhanced in the figures.

A1: Thanks for your suggestion, and western blot data without enhanced in those figures have been replaced as beneath.

Q2. The author concluded that LINC02470 sponges miR-143-3p and SMAD3 consequently activates TGF-β-induced EMT process. Is it possible that knockdown of linc02470 decreases SMAD3 interaction with the promoter or enhancer of the EMT related genes including snail or slug? Could the author show some of ChIP (chromatin immunoprecipitation) data, it would be helpful to understand the model shown in Figure 7A.

A2: Thanks for your suggestion, the data of ChIP for snail and slug have been supplemented.

Reviewer 2 Report

This study first uncovered LINC02470 as a promoter for bladder cancer cell viability, migration, invasion and in vivo tumorigenicity. Mechanistically, this lncRNA acts by sponging miR-143-3p and consequently promoting SMAD3 expression to activate TGF-β induced EMT process. This LINC02470-miR-143-3p-SMAD3 ceRNA axis may be a key determinant in the EMT process underlying bladder cancer progression.

Physical and functional evidence for the existence of LINC02470-miR-143-3p-SMAD3 ceRNA axis remain limited. Some of the concerns that need to be addressed are:

  1. Whether there is a coordinated expression patterns between either two of the components in clinical specimens (either from public or in-house data) should be demonstrated. These correlations would strengthen the ceRNA network hypothesized by the authors.

  1. To demonstrate the functional connection between these components, the authors should perform additional rescuing experiments. For the tumor-related functional experiments shown in this study, only single knockdown/overexpression of LINC02470 was included. Co-treatment experiments should also be performed that SMAD3 is indeed downstream of this regulatory pathway, and that miR-143-3p antagonizes the role of LINC02470.

Author Response

Q1. Whether there is a coordinated expression patterns between either two of the components in clinical specimens (either from public or in-house data) should be demonstrated. These correlations would strengthen the ceRNA network hypothesized by the authors.

A1: Thanks for your suggestion, and the coordinated expression patterns between SMAD3-LINC02470, SMAD3-miR-143 and LINC02470-miR-143 were analyzed with correlation in TCGA-BLCA dataset.

Q2. To demonstrate the functional connection between these components, the authors should perform additional rescuing experiments. For the tumor-related functional experiments shown in this study, only single knockdown/overexpression of LINC02470 was included. Co-treatment experiments should also be performed that SMAD3 is indeed downstream of this regulatory pathway, and that miR-143-3p antagonizes the role of LINC02470.

A2: Thanks for your suggestion and rescuing experiments have been performed as followed.

Reviewer 3 Report

On the basis of previous published data, demonstrating that SMAD3 is a direct target of miR-143-3p and that LNC02470 promotes EMT and aggressiveness of bladder cancer cells,the authors describe the role of LNC02470/miR-143-3p/SMAD3 axis in the EMT process in bladder cancer. Although the  low otiginality/novelty of the manuscript, the  research design is well presented and the data shown sufficiently support the conclusions of the authors. Minor revisionis required; the following comments should be addressed:

-Some revision of English language and style are needed to improve the quality of the manuscript.

-Figure 7a is misleading and needs to be improved: it looks like miR143-3p downregulates the LNC02470 expression rather than being inhibited by LNC02470 sponge actvity.

Author Response

Q1. Some revision of English language and style are needed to improve the quality of the manuscript.

A1: This manuscript has been edit by American Journal Experts Editing.

Q2. Figure 7a is misleading and needs to be improved: it looks like miR143-3p downregulates the LNC02470 expression rather than being inhibited by LNC02470 sponge actvity.

A2: Thanks for your correction, and revised Figure 7a has been replaced as followed.
